# A Preliminary Investigation of the Roles of Endometrial Cells in Endometriosis Development via In Vitro and In Vivo Analyses

**DOI:** 10.3390/ijms25073873

**Published:** 2024-03-30

**Authors:** Yin-Hua Cheng, Ching-Wei Huang, Hao-Ting Lien, Yu-Yang Hsiao, Pei-Ling Weng, Yung-Chiao Chang, Jai-Hong Cheng, Kuo-Chung Lan

**Affiliations:** 1Department of Medical Research and Development, Jen-Ai Hospital, Taichung 412, Taiwan; justjudykimo@gmail.com; 2Department of Obstetrics and Gynecology, Kaohsiung Chang Gung Memorial Hospital, Kaohsiung 833, Taiwan; b9902048@cgmh.org.tw (H.-T.L.); stribte@gmail.com (Y.-Y.H.); lingpay@gmail.com (P.-L.W.); maurenlab15@gmail.com (Y.-C.C.); 3Division of Urology, Department of Surgery, Jen-Ai Hospital, Taichung 412, Taiwan; b9302038@cgmh.org.tw; 4Graduate Institute of Clinical Medical Sciences, Chang Gung University College, Kaohsiung 833, Taiwan; 5Center for Shockwave Medicine and Tissue Engineering, Kaohsiung Chang Gung Memorial Hospital, Chang Gung University College of Medicine, Kaohsiung 833, Taiwan; cjh1106@cgmh.org.tw; 6Medical Research, Kaohsiung Chang Gung Memorial Hospital, Chang Gung University College of Medicine, Kaohsiung 833, Taiwan; 7Department of Leisure and Sports Management, Cheng Shiu University, Kaohsiung 833, Taiwan; 8Department of Obstetrics and Gynecology, Jen-Ai Hospital, Taichung 412, Taiwan; 9Center for Menopause and Reproductive Medicine Research, Kaohsiung Chang Gung Memorial Hospital, Chang Gung University College of Medicine, Kaohsiung 833, Taiwan; 10Department of Obstetrics and Gynecology, Kaohsiung Chang Gung Memorial Hospital, Chang Gung University College of Medicine, Kaohsiung 833, Taiwan

**Keywords:** endometriosis, ECs, LPS, estrogen, NF-κB, VEGF, CK-18, TGF-β, TNF-α

## Abstract

Endometriosis is a complex gynecological disease that affects more than 10% of women in their reproductive years. While surgery can provide temporary relief from women’s pain, symptoms often return in as many as 75% of cases within two years. Previous literature has contributed to theories about the development of endometriosis; however, the exact pathogenesis and etiology remain elusive. We conducted a preliminary investigation into the influence of primary endometrial cells (ECs) on the development and progression of endometriosis. In vitro studies, they were involved in inducing Lipopolysaccharide (LPS) in rat-isolated primary endometrial cells, which resulted in increased nuclear factor-kappa B (NF-κB) and vascular endothelial growth factor (VEGF) mRNA gene expression (quantitative polymerase chain reaction analysis, qPCR) and protein expression (western blot analysis). Additionally, in vivo studies utilized autogenic and allogeneic transplantations (rat to rat) to investigate endometriosis-like lesion cyst size, body weight, protein levels (immunohistochemistry), and mRNA gene expression. These studies demonstrated that estrogen upregulates the gene and protein regulation of cytoskeletal (CK)-18, transforming growth factor-β (TGF-β), VEGF, and tumor necrosis factor (TNF)-α, particularly in the peritoneum. These findings may influence cell proliferation, angiogenesis, fibrosis, and inflammation markers. Consequently, this could exacerbate the occurrence and progression of endometriosis.

## 1. Introduction

Endometriosis is a gynecological disorder caused by estrogen-promoted growth of endometrial cells (ECs) in abnormal locations (ectopic), such as the peritoneum, ovaries, and rectovaginal septum. It is estimated to affect up to 2–17% of females during their reproductive years, leading to significant morbidity [1,2]. The clinical symptoms of endometriosis include chronic pelvic pain, chronic inflammation, dyspareunia, dysmenorrhea, and infertility, which negatively impact the quality of life, productivity, social interactions, and emotional well-being of the patients [3]. Epidemiological studies have indicated an average endometriosis prevalence of 10% in the premenopausal population worldwide [4]. Although surgery offers temporary relief from pain, the symptoms tend to recur in as many as 75% of affected cases within 2 years [5]. Several factors are involved in the development and spread of endometriotic lesions. Several factors contribute to the development of endometriosis, including coelomic metaplasia [6], dysfunctional cellular immunity [7,8,9], genetic factors [10], and a multifactorial mode of inheritance involving specific genes and aberrant environmental factors [11]. However, the exact pathogenesis and etiology of endometriosis remain elusive. Given its complex nature, it is challenging to attribute the pathogenesis of endometriosis to a single factor.

The development of endometriosis is significantly influenced by both cellular inflammation and hormonal factors [12,13,14]. Lipopolysaccharide (LPS), a crucial endotoxin in the outer membrane of gram-negative bacteria, is remarkably stable, serving as a primary biological response modifier and regulating various cell–cell adhesion molecules [15,16]. Moreover, LPS triggers specific symptoms and pathologies associated with various diseases, such as the promotion of cellular inflammatory reactions [17,18] and regulation of cell growth [19,20]. The mRNA and protein expression levels of transforming growth factor (TGF)-β1 are markedly elevated in LPS-induced rabbits, indicating the initiation of epithelial–mesenchymal transition, which leads to severe inflammation and sensitization of endometrial epithelial cells; this process is crucial for fibrotic diseases [21,22]. LPS-induced fibronectin binds to its receptors and facilitates cellular attachment to the peritoneal mesothelium, particularly when ECs are degraded in the pelvis during menstruation [16]. Khan et al. reported that LPS derived from *Escherichia coli* increases colony formation in menstrual blood. They observed higher endotoxin levels in the menstrual fluid and peritoneal fluids (PFs) of women with endometriosis than in the controls. Additionally, protein and mRNA levels of cell proliferation-inducing and pro-inflammatory cytokines, such as hepatocyte growth factor, vascular endothelial cell growth factor (VEGF), interleukin-6 (IL-6), and tumor necrosis factor-α (TNF-α), in the culture media of macrophages are significantly higher in the *E. coli* LPS-treated macrophages than in the untreated macrophages. Furthermore, levels of these molecules are elevated in women with endometriosis compared with those in control women [23]. LPS levels are also elevated in the retained placenta cows, with an average of 2.24 × 10^4^ endotoxin units (EU)/mL, compared with those in the dystocia and healthy postpartum cows [24]. Iba et al. reported that LPS-induced murine endometriosis-like lesions exhibit an increased percentage of Ki67-positive cells and elevated total numbers and sizes of endometrial lesions. Highly expressed LPS-induced fibronectin facilitates the attachment of cells to the peritoneal mesothelium by binding to specific receptors, particularly in the degraded functional ECs of the pelvis during menstruation [16]. mRNA expression levels of VEGF and IL-6 are increased via the nuclear factor (NF)-κB pathway [18]. Although many studies have indicated the additive effects of LPS in the pelvic environment of women, the available information for its application in endometriosis treatment is limited.

Notably, 17β-estradiol (E_2_) is a crucial steroid hormone in the ovaries. It plays key roles in the growth and persistence of endometrial tissue and affects immunomodulation, inflammation, and the pain associated with endometriosis [19,25]. Aberrant estrogen expression contributes to various gynecological diseases, such as endometriosis [26] and endometrial cancer [27], by binding to different estrogen receptors (ERs). These conditions affect many women of reproductive age. Tomio Iwabe et al. revealed that the levels of TNF-α and IL-6 in PF significantly increased with the scores of active-endometriosis lesions, surpassing those observed in patients without endometriosis, via a two-step sandwich enzyme-linked immunosorbent assay. IL-6 levels are elevated in the PF of patients with active red-colored endometriosis, indicating a correlation with endometriosis-associated infertility [28]. Increased production of TGF-β and VEGF promotes inflammation and enhances endometrial angiogenesis in patients with endometriosis, further contributing to endometriosis pathogenesis. Furthermore, NF-κB expression is increased in patients with endometriosis, consistent with reports that NF-κB transcriptional activity is involved in the onset and progression of endometriosis [29,30,31].

Concentrations of LPS and E_2_ vary throughout the menstrual cycle in the pelvis of women with and without endometriosis [19]. Ethical constraints on the continuous monitoring of endometriosis development and progression necessitate the establishment of experimental animal models to understand the molecular mechanisms underlying ectopic endometriosis. Therefore, in this study, we aimed to investigate the involvement and action mechanisms of LPS induction in endometriosis by triggering the secretion of various growth factors and cytokines in isolated primary rat ECs in vitro. Moreover, we explored the differences between two endometriosis rat models in an in vivo study and compared them with the findings of the in vitro study.

## 2. Results

### 2.1. Characterization of Isolated Primary ECs

To understand the progression and pathology of endometriosis, we isolated primary ECs from rat uteri, as previously described [32]. Briefly, ECs were isolated via density gradient centrifugation, revealing a round and small spindle-shaped morphology distinct from that of the uterine endometrial tissue obtained via enzymatic digestion (Figure 1A). The specific region of interest was identified based on the forward and side scattering values in the forward scatter–side scatter dot plot, representing the cell size and granularity, respectively (Figure 1B). A gate (P1) was established to select the primary pool for single-cell events. Fluorescence peaks in the histograms of ECs stained with antibodies against CK-18 and vimentin showed noticeable shifts. These shifts were quantified using flow cytometry to assess the characteristics of ECs (Figure 1C, red curve). Interestingly, 98% of the cells were positive for vimentin, whereas only 16% were positive for CK-18.

### 2.2. Effects of LPS Stimulation on Proliferation, Inflammation, and Fibrosis in Rat ECs In Vitro

We assessed the effects of LPS induction on ECs by determining the expression levels of key markers of EC proliferation, inflammation, and fibrosis using western blotting and RT-qPCR analysis (Figure 1D–O). No significant changes were observed in the protein and mRNA levels of the proliferation-related marker, Ki-67, between the LPS-stimulated ECs and control ECs (0.6 vs. 0.6; *p* = 0.63221; 1.0 vs. 1.1; *p* = 0.7919; Figure 1D,J).

Next, markers associated with fibrosis, such as TGF-β (0.9 vs. 1.0; *p* = 0.4335; 1.0 vs. 1.1; *p* = 0.2872; Figure 1E,K), fibronectin (0.3 vs. 0.3; *p* = 0.1476; 1.0 vs. 1.3; *p* = 0.1725; Figure 1G,M), and vimentin (1.1 vs. 0.9; *p* = 0.2846; 1.0 vs. 0.9; *p* = 0.1108; Figure 1H,N), also showed no significant changes at the protein and mRNA levels. However, in the in vitro study, LPS significantly increased both the protein and mRNA levels of NF-κB (0.8 vs. 1.4; *p* = 0.0374; 1.0 vs. 1.9; *p* < 0.0001; Figure 1F,L) and VEGF (0.6 vs. 1.6; *p* = 0.0364; 1.0 vs. 1.5; *p* = 0.0006; Figure 1I,O) compared with those in the control group. Although LPS stimulation did not affect the proliferation and fibrosis of rat ECs, it promoted inflammation, indicating its potential association with the development of endometriosis symptoms. 

### 2.3. Characteristics and Size of Lesions in the Peritoneal Cavity and Peritoneum of the Rat Endometriosis Models

After observing the cellular inflammatory responses in LPS-induced ECs in vitro, we further investigated whether estrogen influences the molecular and biological mechanisms in rats with endometriosis in vivo via injection of minced uterine tissue and suturing of endometrial fragments. Macroscopic images of endometriotic lesions were implanted into SD rats using an established surgical procedure (Figure 2 and Figure 3). The recipient mice were euthanized four weeks after uterine fragment transplantation and estrogen stimulation, and the intraperitoneal cavity was examined. Cyst formation in endometriosis-like lesions in the peritoneal cavity was confirmed in all rats receiving the uterine fragments. The fluid in the cysts appeared transparent and slightly yellowish (indicated by black circles) compared with that in the control group (Figure 2B). Additionally, the induction of endometriosis-like lesions in the peritoneum of rats by uterine fragment suturing led to severe adhesion and enlargement of the cystic lesions (Figure 3B,C). The mean surface area of the cyst in the endometriosis-like lesion group was approximately 45.1 mm^2^, significantly larger than 13.8 mm^2^ in the control group (on day 28 after estrogen injection; *p* < 0.0001; Figure 3D). However, the mean surface area of the cyst within the peritoneal cavity in the endometriosis-like lesion group was not significantly different from that in the control group (after day 28 of estrogen injection; *p* = 0.0508; Figure 2D). These findings suggest that estrogen promotes the progression of ectopic endometriosis lesions. Moreover, cyst formation is more prominent in the peritoneum than in the peritoneal cavity in ectopic endometriosis-like lesion rat models.

### 2.4. Effect of Estradiol Solution on the Body Weight of Ectopic Endometriosis-like Lesion Rat Model

Next, we determined the safe dose of estradiol solution in vivo in rats with ectopic endometriosis-like conditions. As presented in Table 1, the baseline body weights in the control group were 215.5 ± 6.3 and 227.3 ± 5.8 g. Baseline body weights within the peritoneal cavity and on the peritoneum were 221.6 ± 2.3 and 236.1 ± 3.3 g on day 0, respectively. No significant changes in body weight were observed after the estrogen injection. The control and ectopic endometriosis groups exhibited weights of 206.4 ± 8.2 and 197.3 ± 1.9 g on day 7 (*p* = 0.1244), 215.0 ± 7.3 and 204.4 ± 2.2 g on day 14 (*p* = 0.0949), and 225.7 ± 11.2 and 213.6 ± 4.0 g on day 21 (*p* = 0.2209), and 226.4 ± 14.3 and 207.7 ± 2.0 g on day 28 (*p* = 0.0709), respectively, within the peritoneal cavity. In the other model of ectopic endometriosis, the control and ectopic endometriosis groups exhibited weights of 222.7 ± 8.8 and 231.6 ± 3.7 g on day 7 (*p* = 0.3147), 238.6 ± 9.0 and 242.0 ± 3.9 g on day 14 (*p* = 0.6971), and 252.1 ± 10.9 and 252.4 ± 5.0 g on day 21 (*p* = 0.9782), and 261.5 ± 11.5 and 257.2 ± 5.4 g on day 28 (*p* = 0.7139), respectively, on the peritoneum. Overall, the dosage of estradiol solution used for the development of the two endometriosis-like lesion models did not have any impact on their body weights.

### 2.5. IHC Evaluation of Endometriotic Lesions in Two Ectopic Endometriosis-like Lesion Rat Models

Next, we analyzed ectopic endometriotic tissues to determine the effects of estrogen on rats with endometriosis by using IHC staining. Specifically, we examined the intensity of CK-18, an epithelial marker commonly used in endometriosis studies (Figure 4A and Figure 5A), and quantified the cells stained with epithelial marker CK-18 in both the peritoneal cavity and peritoneum (Figure 4B and Figure 5B). Our investigation revealed an approximately 3-fold increase in the immunoexpression staining of CK-18 in the peritoneum of the ectopic endometrial lesion group compared with that in the control group (16.7% vs. 50.1%; *p* = 0.0303; see Figure 5A,B). However, CK-18 levels in the peritoneal cavity of the endometrial lesion group were not significantly different from those in the control group (4.0 vs. 3.1%; *p* = 0.2911; see Figure 4A,B). Then, expression levels of protein markers, VEGF and TGF-β, were assessed in the peritoneal cavity (Figure 4C,E) and in the peritoneum in the ectopic endometrial lesion (Figure 5C,E). The quantification results demonstrated an approximately 2-fold increase in the expression of VEGF in the ectopic endometrium within the peritoneal cavity and an approximately 1.7-fold increase in the peritoneum compared with those in the control group and an approximately 1.7-fold increase in the peritoneum compared with those in the control group. Furthermore, the protein expression of VEGF was even higher in endometrial lesions on the peritoneum than in the peritoneal cavity (1.5% vs. 3.3%; *p* = 0.0493 and 16.1% vs. 26.9%; *p* = 0.0212; Figure 4D and Figure 5D). A similar trend was observed in the expression of TGF-β in the peritoneal cavity and peritoneum (4.1% vs. 11.6%; *p* = 0.0010 and 18.2% vs. 35.7%; *p* = 0.0302; Figure 4F and Figure 5F). Next, we examined the intensity of TNF-α, a highly potent pro-inflammatory cytokine, in the two ectopic endometrial lesions (Figure 4G and Figure 5G) and quantified the cells stained with TNF-α in both the peritoneal cavity and peritoneum (Figure 4H and Figure 5H). TNF-α IHC staining intensity showed a significant increase of approximately 3-fold within the peritoneal cavity in the ectopic endometrial lesion group compared with that in the control group (1.0% vs. 2.9%; *p* = 0.0004; Figure 4H). This trend was also observed as an approximately 2.2-fold increase in the peritoneum of the ectopic endometrial lesion group, which exhibited a 17.4% positive TNF-α expression compared with the 7.9% (*p* = 0.0387) observed in the control group (Figure 5H).

### 2.6. Effects of Estrogen on Cell Proliferation, Angiogenesis, Fibrosis, and Inflammation-Associated Genes in the Ectopic Endometriosis-like Lesion Rat Models

Next, we investigated the potential mechanisms underlying ectopic endometriosis. We performed uterine tissue mincing and endometrial fragment suturing, followed by estrogen injection, into rat models of endometriosis. As depicted in Figure 6, we evaluated the expression levels of genes associated with proliferation, fibrosis, angiogenesis, and inflammation. *CK-18* levels exhibited a significant increase in both the peritoneal cavity and peritoneum of the ectopic endometrial lesion group, ranging from approximately 2-fold (1.0 vs. 2.3; *p* = 0.0059) to 3-fold (1.2 vs. 3.4; *p* = 0.0082) compared with those in the control group (Figure 6A,B). Gene expression levels of VEGF resulted in approximately a 2-fold increase (1.0 vs. 2.1; *p* = 0.0009; Figure 6C) in the peritoneal cavity and a 5-fold increase (1.2 vs. 6.1; *p* < 0.0001; Figure 6D) in the peritoneum compared with those in the control group. Similarly, gene expression levels of TGF-β were notably elevated following uterine tissue mincing, resulting in approximately a 2-fold increase (1.0 vs. 1.5; *p* = 0.0422; Figure 6E) in the peritoneal cavity. Moreover, suturing of the endometrial fragments led to substantial increases, with a 3-fold rise (1.1 vs. 2.8; *p* = 0.0243; Figure 6F) in levels in the peritoneum of the ectopic endometrial lesion. The gene expression of TNF-α showed approximately a 4-fold increase in the peritoneal cavity-endometriosis rat lesion models and approximately a 142-fold increase in the peritoneum-ectopic endometriosis rat lesion models compared with those in the control group (1.0 vs. 4.2; *p* < 0.0001; 1.0 vs. 141.9; *p* = 0.0008; Figure 6G and Figure 6H, respectively). The mRNA levels of NF-kB were approximately 1-fold (1.0 vs. 1.4; *p* = 0.0091) in the peritoneal cavity-endometrial rat lesion model (Figure 6I), and they significantly increased approximately 20-fold (1.0 vs. 20.9; *p* < 0.0001) in the peritoneum-endometrial rat lesion model (Figure 6J). However, the IL-6 levels exhibited different trends between the peritoneal cavity and the peritoneum, respectively. The mRNA expression of *IL-6* was significantly upregulated (211-fold, 1.5 vs. 317.2; *p* < 0.0001; Figure 6L) in the peritoneum after estrogen injection compared with those in the control group. In contrast, *IL-6* levels in the peritoneal cavity were not significantly different (1.0 vs. 1.2; *p* = 0.0561; Figure 6K) between the control and ectopic endometriosis groups. Interestingly, all of the gene expression of *CK-18*, *VEGF*, *TGF-β*, *TNF*-α, *NF-κB*, and *IL-6* in the endometrial lesions is significantly higher in the peritoneum compared with the peritoneal cavity in these two models.

## 3. Discussion

In this study, we conducted a preliminary investigation of the effects of ECs on the development and progression of endometriosis. LPS induction of rat-isolated primary ECs was performed to investigate the effects on cytokine gene and protein expression levels in vitro. In vivo studies involved autogenic and allogeneic transplantation (from rat to rat) in SD rat models. We created, modified, and compared two rat models simulating ectopic endometriosis and assessed them via transplantation onto the anterior abdominal wall and mesenteric layer to differentiate between rat allografts and autografts. To the best of our knowledge, this study is the first to compare the cyst size of endometriosis-like lesions, body weight, and protein and gene expression levels in the peritoneal cavity and peritoneum between the two rat models and normal controls. Endometriosis is clinically characterized by the presence of endometrial tissue outside the uterine cavity. It involves the aberrant growth and development of the endometrial gland and stroma-associated tissues outside the uterus, indicating its ectopic location. Ectopic endometriotic lesions are occasionally found in other organs or tissues of the body, including the pleural cavity, kidneys, liver, bladder, lungs, gluteal muscles, and brain [33]. The etiology of endometriosis can be explained by metaplasia [34], transplantation [35], or induction [36]. However, these mechanisms do not sufficiently explain the diverse clinical manifestations of endometriosis, given the complexity of the disease and the variable expression, growth, and development of lesions leading to severe disease. Additionally, these mechanisms fail to account for the occurrence of endometriosis in prepubertal girls or newborns.

Various genetic [37,38,39] and immunological factors [40,41,42,43,44], steroid hormones [45,46,47], intrinsic abnormalities of the endometrium [48,49,50,51], and environmental factors [52,53,54,55] collectively influence the development of endometriosis in women. Studies have suggested correlations between endometriosis, chronic inflammation, and cyclic pelvic pain in reproductive-age women [56,57]. In deep endometriosis, pain arises due to the invasion and infiltration of ECs and pro-inflammatory factors into the nerve fibers. These phenomena trigger a disruption in nociceptive modulation, intensifying the neuronal signal toward the somatosensory cerebral cortex [58]. However, the role of inflammasomes in the pathogenesis of endometriosis and the molecular mechanisms underlying the roles of sex hormones in endometriosis-associated pain remain unclear. Etiological studies have reported that infection of the female genital tract by gram-negative bacteria, such as *E. coli* [23,59], *Prevotella* sp. [60], *Fusobacterium necrophorum* [61,62], *Trueprella pyogenes* [63], and various anaerobic species [64], is a significant factor contributing to uterine infection and the initiation of immune responses associated with endometriosis, early abortion, and infertility [65]. The menstrual blood of women with endometriosis exhibits a higher *E. coli* contamination level than that of control women. This contamination is related to the elevated endotoxin levels in both menstrual fluid and PFs [23]. LPS acts as an initial inflammatory mediator derived from bacterial endotoxins and subsequently stimulates immune cells to produce secondary inflammatory mediators, such as cytokines and chemokines. Harada et al. demonstrated that the expression of TNF-α is enhanced in a dose- and time-dependent manner via the activation of NF-κB in *E. coli*-derived LPS (10 ng/mL)-stimulated endometriotic stromal cells in the ovaries of patients with endometriosis [20]. Additionally, intraperitoneal injection of LPS (50 μg/body) increased the mRNA expression of *VEGF* and *IL-6* in mice with surgically induced endometriosis-like lesions via the NF-κB pathway [18]. The transcription factor NF-κB plays a crucial role in regulating innate immunity and inflammation in response to bacteria-derived LPS [66]. In endometriotic lesions, NF-κB is found to be overactive, contributing to the onset, progression, and recurrence of endometriosis. Moreover, NF-κB is known to trigger the production of various pro-inflammatory gene-related chemokines and cytokines, such as TNF-α, IL-6, and IL-8 [67,68]. Many studies have shown that NF-κB modulates inflammation in endometriosis in endometriotic cell line cultures [69,70], endometriotic epithelial cells, and stromal cells from ovarian endometriotic cysts in vitro [71,72,73,74], animal endometriosis models [75,76], and peritoneal endometriotic implants and PFs in vivo [77,78]. Additionally, the p50 and p65 subunits of NF-κB are expressed in human endometrial stromal and epithelial cells [79,80], and they modulate the transcription of numerous genes involved in the regulation of innate immunity, inflammatory response, and cell survival [20,81,82]. Yukihiro et al. reported that the intraperitoneal injection of exogenous LPS (2 mg/kg) twice weekly for four weeks significantly enhanced *VEGF* and *IL-6* mRNA levels and increased the percentage of Ki67-positive endometrial gland epithelia and stromal cells in endometriosis-like implants [18]. VEGF has been verified as a regulator of angiogenesis and neovascularization in women with endometriosis [83,84,85], which are prerequisites for the development of endometriosis [86]. LPS increases VEGF production and promotes angiogenesis in human dental pulp cells through the phosphoinositide 3-kinase, p38, extracellular signal-regulated kinase, c-Jun N-terminal kinase, and NF-κB pathways [87]. These findings suggest that LPS derived from gram-negative bacteria may directly contribute to the inflammation and angiogenesis of endometriosis by influencing the regulation of NF-κB and VEGF in ECs in our in vitro study, although our results showed that Ki-67, TGF-β, fibronectin, and vimentin exhibited no significant changes at both the mRNA and protein levels.

Ki-67 functions as a proliferation marker that specifically targets nuclear proteins expressed exclusively in proliferating cells. Moreover, Ki-67 is expressed in the glandular epithelial cells of the endometrium, with a noticeable presence in the stroma of endometriotic cysts [88]. Nguyen et al. reported a slightly increased median labeling index for Ki-67 in both epithelial and stromal cells of endometriotic cysts in the clinicopathological data of patients with this condition; however, this was not significant compared with the therapeutic group [89]. Vimentin is a marker of endometrial stromal cells and plays a role in coordinating fibroblast proliferation and keratinocyte differentiation during wound healing [90]. Additionally, fibroblasts have been identified as crucial etiological factors in endometriosis [91]. TGF-β serves as a potent growth factor and monocyte chemoattractant with diverse biological functions [92]. It regulates various cellular processes crucial for endometriosis lesion development, including cell adhesion [93], invasion [94], inflammation [95,96], fibrosis [97], and angiogenesis [98]. LPS significantly increased the mRNA and protein expression of TGF-β and VEGF in ectopic endometrial stromal cells isolated from patients with adenomyosis, a specific type of endometriosis. This suggests that LPS induces the inflammatory, proliferative, and invasive growth progression of adenomyosis [99]. Resveratrol, a nonflavonoid polyphenol, has been reported to inhibit NF-κB to regulate the expression of several genes and cytokines, such as the profibrogenic factor TGF-β [100]. Fibronectin, a cell–cell adhesion molecule, is highly expressed in both glands and stromal cells. Khaleque et al. reported that the LPS-induced overexpression of fibronectin facilitates cellular attachment to the peritoneal mesothelium after binding to specific receptors. This phenomenon occurs when degraded functional ECs appear in the pelvis during the menstrual period [16]. Importantly, LPS-associated molecular patterns are recognized by the pattern recognition receptors (PRRs) in the female reproductive tract. These receptors are expressed in various mammalian cells, including macrophages, dendritic cells, neutrophils, natural killer cells, and epithelial cells of the innate immune system. Toll-like receptors are a group of PRRs [101,102,103]. In our in vitro study, LPS stimulation did not affect EC proliferation or fibrosis but promoted inflammation in ECs. This suggests a direct role of LPS stimulation in mediating inflammation and angiogenesis in isolated rat primary ECs. However, these results do not indicate the specific cell type involved in the regulation of molecular mechanisms in endometriosis.

To better understand the development and progression of endometriosis in vivo and better mimic patients with endometriosis in a clinical setting, we developed and modified two ectopic endometriosis-like lesion rat models following previously reported methodologies. Owing to the absence of uterine shedding (menses) in rats, the initiation of uterine remodeling to prevent retrograde menstruation cannot fully replicate the actual endometriosis phenomenon observed in human clinical settings. Two methods are commonly employed for this purpose. The first involves pooling individual endometrial-minced tissues from a donor dispersed into the peritoneal cavity of a recipient through intraperitoneal, which has demonstrated increased macrophage recruitment and production of inflammatory cytokines [104,105]. The second method, as indicated in the existing literature, frequently involves inducing endometriosis in mice through direct suturing of organized endometrial fragments or artificially decidualized endometrial tissue into the peritoneal cavity or peritoneum [106,107,108]. Numerous rodent and non-human primate studies have established that endometrial tissue fragments and non-dissociated epithelia, stroma, and glands can initiate endometriosis, which resembles the process of retrograde menstruation [109,110,111]. Although these two models utilize estrogen injections, there is still a lack of comparative research evaluating the severity of endometriosis in these rat models. Additionally, there is limited discussion regarding the similarities or differences in the molecular mechanisms between the two.

Estrogen is a causative factor in endometriosis. Its levels are elevated in the menstrual blood of women with endometriosis [112,113], suggesting that estrogen is formed locally in the endometrium of patients with endometriosis [114]. Several estrogen-metabolizing enzymes, such as aromatase and 17β-hydroxysteroid dehydrogenase-1, 2, 5, 7, and 12, are aberrantly expressed, leading to high estrogen biosynthesis and low estrogen inactivation. This aberration results in excess local estrogen in the ectopic endometrium, causing proliferation of the ectopic endometrium and affecting gene and protein expression in a paracrine manner within the local microenvironment of the ectopic endometrium [4,115]. These changes contribute to the development of different types of endometriosis, including ovarian, peritoneal, and deep infiltrating endometriosis [113,116,117,118,119,120,121]. In addition to the impact of estrogen and its related enzymes on the progression of endometriosis, the levels of ERα and ERβ are affected in the ectopic endometrium of women with endometriosis [122,123]. ERα and ERβ levels exhibit opposite trends and perform antagonistic functions in endometriotic tissues [112,122,124]. The *ERα* gene is necessary for normal uterine and ectopic lesion development in *ERα* and *ERβ* knockout mice [125]. The *ERβ* gene plays a role in modulating the inflammasome and apoptosis complexes in the pathogenesis of endometriosis [124,126]. Additionally, peritoneal macrophages in women with endometriosis exhibit an overexpression of both ERα and ERβ [127], along with elevated levels of inflammatory cytokines, including VEGF, IL-6, and TNF-α [128].

Here, estrogen injections did not have significant effects on the body weights of the two endometriosis-like lesion rat models (Table 1). However, they did lead to an increase in the cyst size within endometriosis-like lesions, particularly in the peritoneum, in an ectopic endometriosis-like lesion rat model (Figure 3). This suggests that the estrogen dose is safe and may potentially contribute to the study of the molecular mechanisms of ectopic endometriosis following its characterization. In our subsequent analysis, we employed immunohistochemistry procedures to assess CK-18, an epithelial cell marker, to confirm their endometrial characterization. CK-18 is a recognized and valuable marker for endometriosis studies [129]. In our study, the protein expression of the CK-18-positive area and the mRNA expression of *CK-18* consistently increased in the endometriosis group compared with the control group in the peritoneum, as depicted in Figure 5B and Figure 6B. However, in the other endometriosis model where uterine fragments were dispersed within the peritoneal cavity, the investigation revealed that the protein expression in the CK-18-positive area remained unaltered, while there was an increase in the mRNA expression of *CK-18* between the endometriosis and control groups (Figure 4B and Figure 6A). This implies that estrogen plays a role in promoting the regeneration of endometrial epithelial cells identified by CK-18 in endometriotic lesions in the peritoneum through suturing. Moreover, the peritoneal symptoms exhibited by the endometriosis group were stronger than the peritoneal cavity symptoms. However, it is important to note that the endometriosis group within the peritoneal cavity may engage in alternative signaling pathways for the regeneration of ECs other than CK-18. We examined VEGF, recognized as the most potent angiogenic factor, using immunohistochemistry and qPCR analysis to compare its levels between the two rat ectopic endometriosis models. We have investigated that the protein expression of the VEGF-positive area and the mRNA expression of *VEGF* increased in the endometriosis group compared with the control group within the peritoneal cavity, as depicted in Figure 4D and Figure 6C. Similarly, trends were observed in the protein expression of the VEGF-positive area and the mRNA expression of *VEGF* in the peritoneum, as depicted in Figure 5D and Figure 6D. Li et al. suggest that VEGF-C is released by pro-inflammatory cytokine-stimulated endometriotic stromal cells. This upregulation contributes to enhanced lymphatic vessel infiltration into the endometriotic lesions, thereby promoting increased lymphangiogenesis. This process is a crucial modulator of endometriosis progression by stimulating lymphangiogenesis [106]. RNA levels of *VEGF* are elevated in the normal endometrium during the secretory phase of the menstrual cycle [130,131]. Furthermore, estradiol upregulates *VEGF* gene expression in normal human endometrial tissues [131], endometrial carcinoma cell lines [132], and rodent uteri in vivo [133]. Therefore, angiogenesis is crucial for the growth and survival of endometriotic lesions.

TGF-β levels are notably elevated in the PF, serum, ectopic endometrium, and peritoneum of women with endometriosis compared with those without endometriosis. Increased levels of TGF-β significantly contribute to the progression of endometriosis [134]. Our study indicated that the protein expression of the TGF-β-positive area and the mRNA levels of TGF-β increased in the endometriosis group compared with the control group within the peritoneal cavity, as depicted in Figure 4F and Figure 6E. Similarly, trends were observed in the protein expression of the TGF-β-positive area and the mRNA expression of *TGF-β* in the peritoneum, as depicted in Figure 5F and Figure 6F. Elevated estradiol levels lead to the activation of peritoneal macrophages and subsequent inflammation in the abdominal cavity [135]. These results suggest estrogen may induce ECs to participate in fibrosis and/or angiogenesis by increasing the mRNA and protein expression levels of TGF-β.

TNF-α is produced by neutrophils, activated macrophages, lymphocytes, and several non-hematopoietic cells. Its major function is to initiate the factors associated with inflammatory responses and cytokine cascades [44]. TNF-α levels are elevated in the PFs of both patients and rat models of endometriosis [136,137,138]. In this study, both the TNF-α positive area and mRNA levels of *TNF-α* were elevated in the endometriosis group compared with those in the control group. These results revealed that not only the protein expression of TNF-α increased in the endometriotic lesions within the peritoneum but also in the peritoneal cavity (Figure 4G,H and Figure 5G,H). Additionally, the endometriotic lesions within the peritoneum exhibited stronger staining intensity than those within the peritoneal cavity. We further investigated whether the elevation in TNF-α levels is accompanied by the regulation of upstream transcription factors. NF-κB, a transcription factor, plays crucial roles in the immune and inflammatory responses, modulating cell proliferation, apoptosis, adhesion, invasion, and angiogenesis in various cell types [139,140]. These cellular processes are implicated in the early development of endometriotic lesions in vivo and in the overall development of endometriosis [4,28], further regulating genes associated with inflammation, including *IL-6* and other pro-inflammatory cytokines [141,142,143]. Our results indicate that estrogen indirectly induces immune cells to participate in inflammatory effects by regulating the mRNA expression levels of *NF-κB*, *TNF-α*, and *IL-6*, particularly in the ectopic endometrial lesion model in the peritoneum. However, gene expression alone is not sufficient to fully explain the molecular mechanisms and signaling pathways associated with endometriosis. We aim to conduct further comprehensive experiments to understand these mechanisms. Future studies should isolate purified cells to determine whether specific cells play a role in the molecular mechanisms of endometriosis. Here, we investigated the involvement of estrogen and immune cells in the pathology and development of endometriosis in two rat models. Further analysis of secreted cytokines from endometriotic lesions is essential to providing valuable insights into the pathogenesis and pathology of endometriosis.

## 4. Materials and Methods

### 4.1. Animals

Female Sprague–Dawley (SD) rats (age: 7–8 weeks; body weight: 200–250 g) obtained from BioLASCO (Taipei, Taiwan) were used in this study. The rats were randomly transferred to plastic cages filled with aspen bedding, with three rats per cage. They were acclimatized for one week before the experiments. They were kept under a 12/12 h light/dark cycle with controlled temperature (23 ± 1.5 °C) and humidity (relative humidity 40–60%). All experimental animals were handled as per the specifications of the Animal Center of Kaohsiung Chang Gung Memorial Hospital (IACUC no. 2022120101; approval date: 20230119) and relevant guidelines provided by the National Institutes of Health (NIH) for the ethical use and treatment of laboratory animals.

### 4.2. Preparation of Primary Rat Uterine ECs

Primary rat uterine ECs were isolated as described by De Clercq et al. [32], with slight modifications. To stimulate the proliferation of ECs for increased yield, E_2_ solution (E1024; Sigma) at 40 μg/kg was subcutaneously administered to rats for three consecutive days prior to euthanasia. Animals were euthanized under CO_2_-induced anesthesia, followed by disinfection with 75% ethanol. The abdominal cavity was opened, and uterine horns were exposed by clipping and removing the fat and connective tissue. The uterine horns were then incised to expose the uterine cavity and washed with 1× phosphate-buffered saline (PBS; 10010-023; Gibco, Life Science, Grand Island, NY, USA). Subsequently, the uterine horns were cut into small fragments and transferred to 15-mL tubes containing a solution of 0.1% collagenase (S1746501; Nordmark, Uetersen, Germany) in 0.25% trypsin-EDTA (25200-072; Invitrogen, Carlsbad, CA, USA). The mixture was incubated at 37 °C and 1200 rpm with vigorous shaking every 60 min (Thermomixer Comfort; Eppendorf AG, Hamburg, Germany). The digested suspension of primary rat ECs was filtered through a 70-μm nylon mesh and centrifuged at 1000 rpm for 5 min. Primary rat ECs were cultured in Dulbecco’s modified Eagle’s medium/nutrient mixture F-12 (Thermo Fisher Scientific, Grand Island, NY, USA) supplemented with 10% fetal bovine serum (FBS; 10437028; Invitrogen, CA, USA), 100 U/mL penicillin/streptomycin (15140-122; Invitrogen, CA, USA), and 2 mM l-glutamine (25030-081; Invitrogen, CA, USA) in a humidified incubator (Forma Series II 3110 Water-Jacketed CO_2_ Incubator; Thermo Fisher Scientific, Waltham, MA, USA) at 37 °C with 5% CO_2_. After reaching approximately 90% confluency in the first passage, primary rat ECs were harvested for subsequent experiments.

### 4.3. Flow Cytometric Analysis of ECs

After culturing ECs until the second passage, cytokeratin 18 (CK-18) and vimentin were used to identify the purity of the epithelial and stromal cells by analyzing them using cell surface markers via flow cytometry. Briefly, ECs were incubated with CK-18 (bs-2043R-A488; BIOSS, Woburn, MA, USA) and vimentin (ab92547; Abcam, Cambridge, UK) for 30 min on ice. After washing with the FACS wash buffer (PBS containing 10% FBS), ECs were incubated with the secondary goat-anti-rabbit IgG Alexa Fluor 750 antibody on ice for 20 min and analyzed using FACSAria II (BD Biosciences, San Jose, CA, USA).

### 4.4. LPS Stimulation of Primary Rat Uterine ECs

Rat ECs were seeded into 10-cm plastic culture dishes at a density of 5 × 10^5^ cells per plate for the experiments. The cells were stimulated with 100 ng/mL LPS (L2630; Sigma-Aldrich, Louis, MO, USA) for 48 h to induce inflammation. Then, LPS was removed, fresh medium was added, and incubated for 1 h for subsequent experiments.

### 4.5. Establishment of Two Rat Endometriosis Models

Experimental procedures and tissue collection:

All rats underwent surgery under anesthesia, with zoletil 50 (20–40 mg/kg; Virbac, Carros, France) and xylazine (5–10 mg/kg; Rompun, Bayer AG, Leverkusen, Germany) administered via intraperitoneal injection. Once anesthetized, the rats were positioned supine, their lower abdomens were shaved, and the area was disinfected with iodine solution and 75% ethanol. Two procedures were used to develop ectopic endometriosis models:

Procedure 1:

Procedure 1 followed the method outlined by Azuma et al. [18], with slight modifications. After anesthetization, a minor midline incision (approximately 0.5 cm) was made below the abdomen in the recipient rats. The whole uterus from the donor rat was excised, and any excess tissue was rinsed off using sterile saline. The whole uterus was then longitudinally cut and finely minced (approximately 0.3 mm diameter) using dissecting scissors. Minced donor tissue (endometrial tissue) was suspended in 500 μL of 0.9% normal saline and dispersed into the peritoneal cavity of recipient rats. The peritoneum was sutured (at a donor-to-host ratio of 1:2). Afterward, E_2_ solution (E1024; Sigma-Aldrich, Louis, MO, USA) at a concentration of 2.5 mg/kg/day or normal saline was administered for four consecutive weeks, starting from the day after the operation [144]. After four weeks, the peritoneal cavities of the rat were meticulously examined, the endometriotic lesions were photographed, and their sizes were recorded. Then endometriosis-like lesions were delicately excised for subsequent studies. The detailed protocol is shown in Figure 2A.

Procedure 2:

Under Procedure 2, surgery was performed as described by Zhanfei et al. [145], with slight modifications. Following the induction of anesthesia in recipient rats, the abdominal skin was shaved, and a roughly 5-cm incision was made along the midline of the abdomen to access the abdominal cavity. The left uterine horn was ligated, excised, and immersed in 0.9% normal saline solution. The remaining tissue was removed, and the uterus was sectioned to a size of 5 × 5 mm^2^. Subsequently, the incised uterine fragments were sutured to the inner side of the abdominal wall, with the endometrial fragments facing toward the peritoneal cavity. Finally, abdominal muscles and skin were sutured. E_2_ (2.5 mg/kg/day) or normal saline was administered for four weeks to induce endometriosis; the endometriotic lesions were photographed, and their sizes were recorded. Then, endometriosis-like lesions were delicately excised after the operation for subsequent studies. The detailed protocol is shown in Figure 3A. Endometriosis-like lesions on the peritoneum of rats were collected for further studies.

### 4.6. RNA Extraction, Reverse Transcription (RT), and Quantitative Polymerase Chain Reaction (qPCR)

Total RNA was isolated using the Quick-RNA Miniprep Kit (Zymo Research, Irvine, CA, USA), following the manufacturer’s instructions. The isolated RNA was reverse-transcribed into cDNA and subjected to qRT-PCR using the Fast SYBR Green Master Mix (Applied Biosystems, Vilnius, Lithuania) with the ABI 7500 Fast Real-Time PCR System (Applied Biosystems, Singapore). β-actin served as an internal RNA control, and each sample was normalized based on its β-actin levels. All primer sequences used in this study are listed in Table 2.

### 4.7. Western Blotting

Frozen EC samples were homogenized by incubating ECs with 200 μL of T-Pro-RIPA-Lysis buffer (JT 89-L001M; T-Pro Biotechnology, Taipei, Taiwan) on ice for 30–60 min. Subsequently, the samples were centrifuged at 14,000 rpm for 15 min at 4 °C. The collected supernatant was loaded with 10 μg protein onto a 10% sodium dodecyl sulfate-polyacrylamide gel with a 5% stacking gel and subjected to electrophoresis. Then, proteins were transferred onto a polyvinylidene fluoride membrane (IPVH85R; Millipore, Billerica, MA, USA). To block the non-specific binding sites, a solution of 5% skim milk powder in TBS-T was applied at room temperature for 60 min. Primary antibodies, including Ki-67 (SAB5700770; Sigma-Aldrich, Louis, MO, USA), TGF-β (#3711; Cell Signaling Technology, Beverly, MA, USA), total NF-κB (#8242; Cell Signaling Technology, MA, USA), phospho-NF-κB (#3033; Cell Signaling Technology, MA, USA), fibronectin (bs-006R; BIOSS, Woburn, MA, USA), vimentin (ab92547; Abcam, Cambridge, UK), VEGF (GTX102643; GeneTex, Irvine, CA), and GAPDH (MA5-15738; Invitrogen, CA, USA), were incubated with the membrane at 4 °C overnight. Then, the membrane was incubated with anti-mouse-IgG horseradish peroxidase (HRP; AP124P; Millipore, MA, USA) or anti-rabbit-IgG HRP (A0545; Sigma-Aldrich, Louis, MO, USA) secondary antibodies at room temperature for 60 min and analyzed using the Immobilon Western Chemiluminescent HRP Substrate (WBKLS0500; Millipore, MA, USA) for Dot/Slot Blot. Images were captured using an electronic image analyzer (BIO-RAD ChemiDOC MP system, Hercules, CA, USA), and signal intensity was quantified using VisionWorks^®^8.18 software (Analytik Jena, Jena, Germany).

### 4.8. Immunohistochemical (IHC) Staining

Uterine sections were prepared as previously described [156]. Briefly, slides were dried at 55 °C for 30 min, deparaffinized, cleaned using the Sub-X Cleaning Medium (Leria SUB-X 3803670, Carterville, IL, USA), and hydrated in various concentrations of ethanol. Antigen retrieval was performed using citrate buffer (K-035; Diagnostic Biosystems, København, Denmark) at pH 6.0 via autoclave heat treatment. After inhibiting endogenous peroxidase (ab64218; Abcam, Cambridge, UK) for 15 min at room temperature, slides were incubated with TGF-β (bs-4538R; BIOSS, Massachusetts, USA) at 4 °C overnight. The secondary antibody was applied using DAKO REAL^TM^ EnVision (DAKO K5007, Silkeborg, Denmark) and incubated for 30 min at room temperature, followed by Rabbit/Mouse (DAB^+^) incubation for 5 min. The slides were counterstained with hematoxylin (105175; Merck, Millipore, MA, USA) for 10 min and mounted using CC/Mount (C9368; Sigma-Aldrich, Louis MO, USA). Throughout the experiments, thorough washing was performed with TBS-T. Scanning was conducted using the panoramic scanner (Sysmex Europe GmbH, Norderstedt, Germany), and images were captured using a microscope (Sysmex TOA Medical Electronics, Europe) GmbH, Hamburg, Germany). Finally, results were quantified using the ImageJ software version 1.8.0. (National Institutes of Health, Bethesda, ML, USA).

### 4.9. Statistical Analyses

For homogeneous data, unpaired, two-tailed Student *t* tests were used to evaluate the statistical differences between the LPS-treated/estrogen-injected endometriosis and control groups. All analyses were conducted using the Prism software (version 5.0; GraphPad Software Inc., San Diego, CA, USA). Statistical significance was set at *p* < 0.05. Results are presented as the mean ± standard error.

## 5. Conclusions

In this study, we investigated the effects of LPS and 17β-estrogen on the development and progression of endometriosis through in vitro and in vivo studies. Our in vitro research revealed an increase in NF-κB and VEGF mRNA gene expression, as well as protein expression. Furthermore, our in vivo experiments demonstrated that estrogen upregulates the gene and protein expression of CK-18, TGF-β, VEGF, and TNF-α, particularly in the peritoneum. Although the molecular mechanisms and associated signaling pathways were not entirely identical, they significantly influenced EC proliferation, angiogenesis, fibrosis, and inflammation in endometriosis models, particularly those with peritoneal involvement. In the future, we will investigate the specific cell types involved in peritoneum-ectopic endometriosis, such as macrophages and dendritic cells, and how to regulate the expression of cytokine genes and their inflammatory effects. Additionally, it is crucial to understand estrogen biosynthesis by aromatase or 17β-hydroxysteroid dehydrogenase in the local environment of ectopic endometriosis to identify potential candidate inhibitors for treating this characterization of endometriosis, further strengthening our findings to contribute to a deeper understanding of ectopic endometriosis.

## Figures and Tables

**Figure 1 ijms-25-03873-f001:**
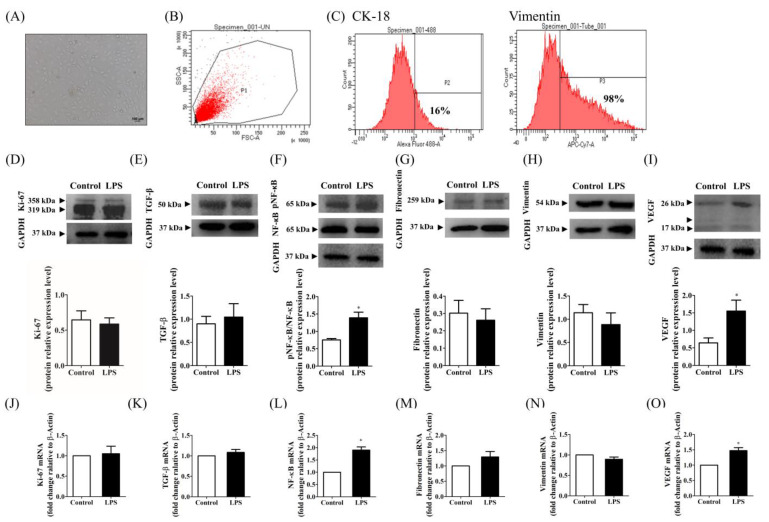
In vitro studies to isolate primary endometrial cells (ECs) and determine the protein and mRNA expression levels in lipopolysaccharide (LPS)-induced rat primary ECs and control ECs. (**A**) Images of rat primary ECs captured seven days after isolation, exhibiting a diameter of 100 μm. (**B**,**C**) Flow cytometry analysis revealed representative histograms of EC markers, indicating the percentages of positive staining for CK−18 and vimentin. Supernatants were collected from primary ECs (3 × 10^3^) for western blotting and real-time polymerase chain reaction (PCR) analysis to determine the expression levels of Ki−67 (**D**,**J**), transforming growth factor (TGF)-β (**E**,**K**), nuclear factor (NF)-κB (**F**,**L**), fibronectin (**G**,**M**), vimentin (**H**,**N**), and vascular endothelial cell growth factor (VEGF) (**I**,**O**) in rat primary ECs after 48 h of LPS stimulation. Bar graphs represent the standard error of the mean (SE) of 3–6 samples pooled from the experiment. * *p* < 0.05, LPS-induced ECs vs. control ECs. TGF-β, transforming growth factor-β; NF-κB, nuclear factor-kappa B; VEGF, vascular endothelial growth factor. Unpaired, two-tailed Student *t* tests were used to evaluate the statistical differences between (LPS)-induced rat primary ECs and control ECs.

**Figure 2 ijms-25-03873-f002:**
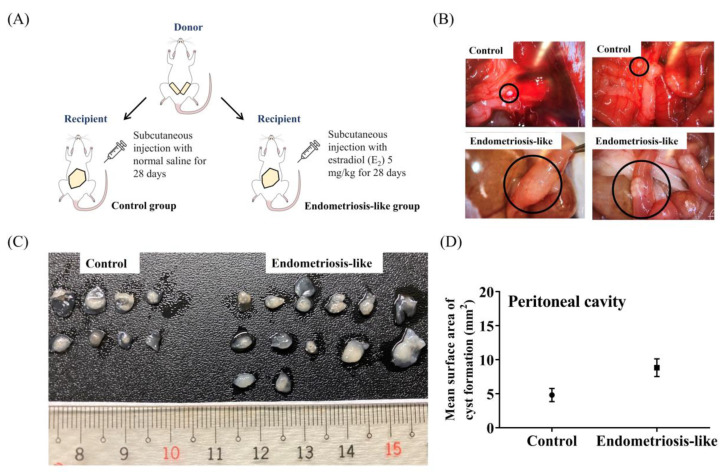
Investigation and characterization of the peritoneal cavity in the endometriosis-like lesion rat model. (**A**) A schematic diagram of the experimental setup to establish a rat model with endometriosis in the peritoneal cavity. (**B**) Representative images of the endometriotic tissues in the peritoneal cavity were recorded before the sacrifice. The upper panel displays the normal peritoneal cavity of a control rat. The lower panel exhibits the presence of endometriosis-like lesions and adhesion of the gastrointestinal tract. The endometriosis-like lesions are indicated by black circles. (**C**) Comparison of the number and size of endometriosis-like lesions on the right side with those on the left side of the control group. (**D**) Comparison of the cyst size of endometriosis-like lesions in the peritoneal cavity with that in the normal control group in vivo. The data are expressed as the mean ± SE of 7–13 samples pooled from the experiments. Unpaired, two-tailed Student *t* tests were used to evaluate the statistical differences between ectopic ECs in the peritoneal cavity of the endometriosis group compared with those in the control group.

**Figure 3 ijms-25-03873-f003:**
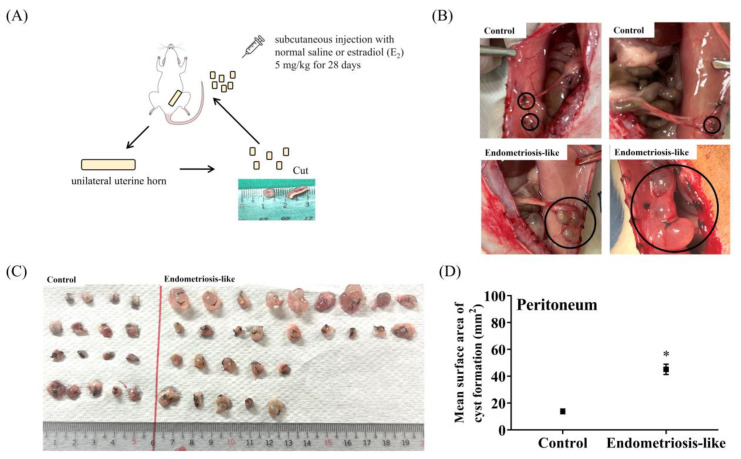
Establishment, characterization, and analysis of the peritoneum in an ectopic endometriosis-like lesion rat model. (**A**) A schematic diagram illustrates the experimental setup used to establish a rat model with ectopic endometriosis in the peritoneum. (**B**) Representative images of the endometriotic tissues in the peritoneum were recorded before the sacrifice. The upper panel shows the normal peritoneum of a control rat, and the lower panel indicates the presence of endometriosis-like lesions and adhesions in the peritoneum. (The endometriosis-like lesions are highlighted by black circles.) (**C**) Comparison of the number and size of endometriosis-like lesions on the right side with those on the left side of the control group. (**D**) Comparison of the cyst size of endometriosis-like lesions in the peritoneum compared with that in the normal control group. The data are expressed as the mean ± SE of 16–26 samples pooled from the experiments. Unpaired, two-tailed Student *t* tests were used to evaluate the statistical differences between ectopic ECs in the peritoneal cavity of the endometriosis group compared with those in the control group. * *p* < 0.05 was significant compared with the control group.

**Figure 4 ijms-25-03873-f004:**
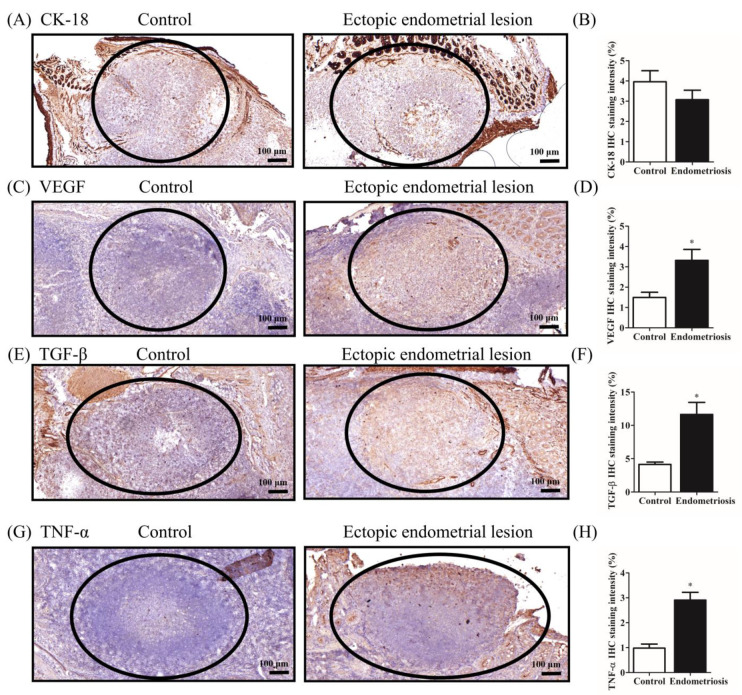
Immunohistochemical evaluation of the ectopic endometrium of a rat model of endometriosis in the peritoneal cavity. (**A**,**C**,**E**,**G**) Immunohistochemical analysis of cytokeratin (CK)-18, VEGF, TGF-β, and tumor necrosis factor (TNF)-α in the endometrium of the control rat (**left**) and ectopic endometrial tissues (**right**) day 28 post-surgery (brown staining). The black circle indicates the protein-positive signaling for the primary antibody (bar: 100 μm). (**B**,**D**,**F**,**H**) Analysis of CK-18, VEGF, TGF-β, and TNF-α-positive areas based on immunohistochemistry results. Data are expressed as the mean ± SE. * *p* < 0.05 for Student *t* tests (unpaired, two-tailed). (*n* ≥ 3) to evaluate the statistical differences between the estrogen-injected endometriosis and control groups. CK-18, cytokeratin 18; VEGF, vascular endothelial growth factor; TGF-β, transforming growth factor β; TNF-α, tumor necrosis factor-α.

**Figure 5 ijms-25-03873-f005:**
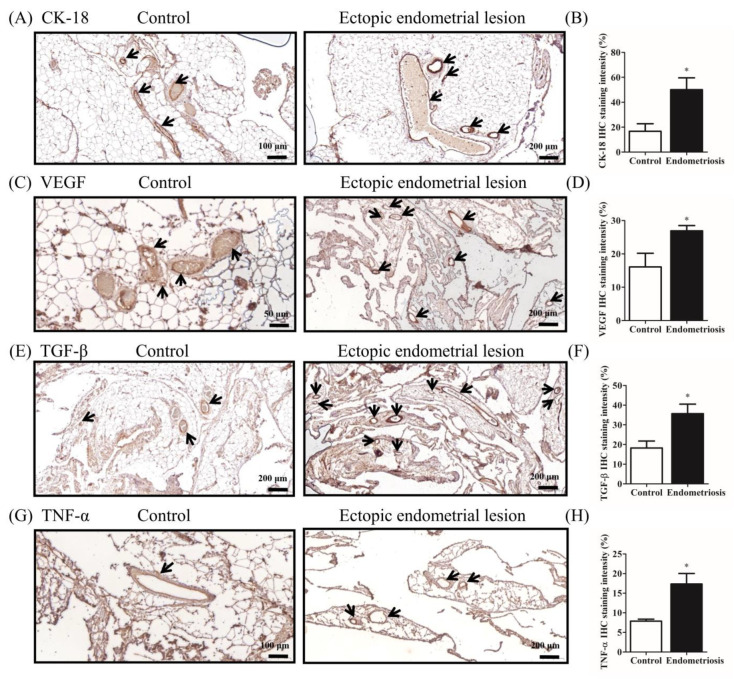
(**A**,**C**,**E**,**G**) Immunohistochemical analysis of CK-18, VEGF, TGF-β, and TNF-α expression levels in immunohistochemically stained ectopic endometrial lesions in the peritoneum on day 28 after estrogen injection. Black arrows indicate the protein-positive expression for the primary antibody (bar: 50–200 μm). (**B**,**D**,**F**,**H**) Immunohistochemical staining intensity of CK-18, VEGF, TGF-β, and TNF-α-positive areas based on immunohistochemistry results. Data are expressed as the mean ± SE. * *p* < 0.05 for Student *t* tests (unpaired, two-tailed). (*n* ≥ 3) difference was significant compared with the control group. CK-18, cytokeratin 18; VEGF, vascular endothelial growth factor; TGF-β, transforming growth factor β; TNF-α, tumor necrosis factor-α.

**Figure 6 ijms-25-03873-f006:**
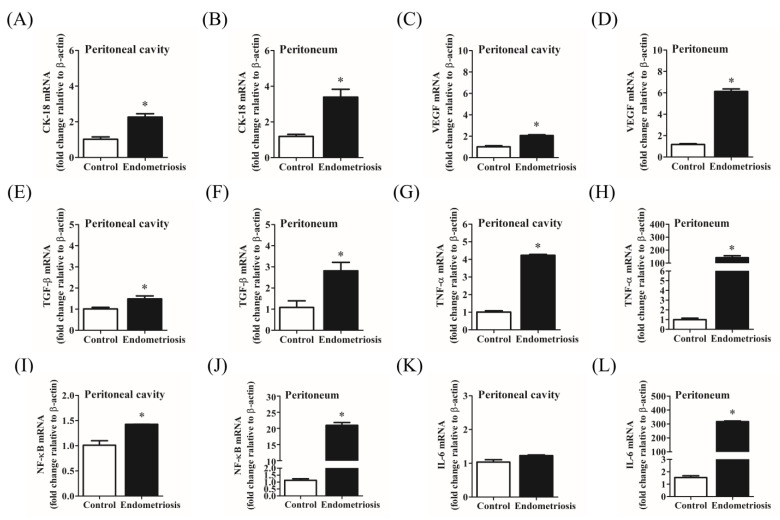
Expression levels of *CK-18*, *VEGF*, *TGF-β*, *TNF-α*, *NF-κB*, and *IL-6* in the ectopic endometrial lesions of the two rat models of endometriosis. Panels illustrate the mRNA levels of *CK-18* (**A**,**B**), *VEGF* (**C**,**D**), *TGF-β* (**E**,**F**), *TNF-α* (**G**,**H**), *NF-κB* (**I**,**J**), and *IL-6* (**K**,**L**) in the ectopic endometrial lesions of the peritoneal cavity and peritoneum compared with those in the control group on day 28 after estrogen injection. All values were normalized to β-actin. Data are expressed as the mean ± SE. * *p* < 0.05 for Student *t* tests (unpaired, two-tailed). (*n* ≥ 3) to evaluate the statistical differences between the estrogen-injected endometriosis and control groups. CK-18: cytokeratin 18; VEGF: vascular endothelial growth factor; TGF-β: transforming growth factor β; TNF-α: tumor necrosis factor-α; NF-κB: nuclear factor-kappa B; IL-6: interleukin-6.

**Table 1 ijms-25-03873-t001:** Estradiol solution does not exert any discernible effects on the body weights (g) of the two ectopic endometriosis-like lesion rat models.

	Control ^a^	Ectopic Endometrial Lesion in the Peritoneal Cavity (Procedure 1)	*p* Value	Control ^a^	Ectopic Endometrial Lesion in the Peritoneum (Procedure 2)	*p* Value
Day 0	215.5 ± 6.3	221.6 ± 2.3	0.2641	227.3 ± 5.8	236.1 ± 3.3	0.2382
Day 7	206.4 ± 8.2	197.3 ± 1.9	0.1244	222.7 ± 8.8	231.6 ± 3.7	0.3147
Day 14	215.0 ± 7.3	204.4 ± 2.2	0.0949	238.6 ± 9.0	242.0 ± 3.9	0.6971
Day 21	225.7 ± 11.2	213.6 ± 4.0	0.2209	252.1 ± 10.9	252.4 ± 5.0	0.9782
Day 28	226.4 ± 14.3	207.7 ± 2.0	0.0709	261.5 ± 11.5	257.2 ± 5.4	0.7139

Notes: ^a^ Control, subcutaneous injection with normal saline. The data are expressed as the mean ± SE of 3–14 samples pooled from the experiments. Unpaired, two-tailed Student *t* tests were used to evaluate the statistical differences between the estrogen-injected endometriosis and control groups.

**Table 2 ijms-25-03873-t002:** Primers of specific genes used for the quantitative reverse transcription-polymerase chain reaction (qRT-PCR) analysis of endometrial cells and rat specimens.

Gene	Forward Primer (5′–3′)	Reverse Primer (5′–3′)
*Ki-67* [146]	CTGCAGAGAAGGTTGGGATAAA	CTGACTTTGCCCAGAGATGAA
*TGF-β* [147]	TAATGGTGGACCGCAACAACG	GGCACTGCTTCCCGAATGTCT
*NF-κB* [148]	CTGGCAGCTCTTCTCAAAGC	CCAGGTCATAGAGAGGCTCAA
*Fibronectin* [149]	GACTCGCTTTGACTTCACCAC	GCTGAGACCCAGGAGACCAC
*Vimentin* [150]	GCACCCTGCAGTCATTCAGA	GCAAGGATTCCACTTTACGTTCA
*VEGF* [151]	TATCTTCAAGCCGTCCTGTG	GATCCGCATGATCTGCATAG
*CK-18* [152]	CTGGGGCCACTACTTCAAGA	CCTTGCGGAGTCCATGAATG
*IL-6* [153]	TCAACTCCATCTGCCCTTCAG	AAGGCAACTGGCTGGAAGTCT
*TNF-α* [154]	GCCTCTTCTCATTCCTGCTT	CACTTGGTGGTTTGCTACGA
*β-actin* [155]	GACGTTGACATCCGTAAAGACC	CTAGGAGCCAGGGCAGTAATCT

## Data Availability

The data generated/analyzed in this study are available from the corresponding author upon request.

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
