# Peer review of "A Preliminary Investigation of the Roles of Endometrial Cells in Endometriosis Development via In Vitro and In Vivo Analyses"

_ijms, 2024, doi:10.3390/ijms25073873_

Round 1

Reviewer 1 Report

Comments and Suggestions for Authors

A Preliminary Investigation of the Roles of Endometrial Cells in Endometriosis Development via In Vitro and In Vivo Analyses

In this study, the authors conducted a preliminary investigation into the influence of primary endometrial cells (ECs) on the development and progression of endometriosis. The study was well thought out and executed but some major points must be addressed. I suggest a new evaluation after the corrections.

Introduction

1)    Line 53: The epidemiological data is misleading once the data here presented, is from adolescents with dysmenorrhea. Please, correct this information.

2)    Line 56-57: Reference is missing.

3)    Line 59: Please, add the reference DOI: 10.1007/s00404-021-05971-6 in the sentence “Although surgery offers temporary relief from pain, the symptoms tend to recur in as many as 75% of affected cases within 2 years.”

Figures

4)    Figure 1: Please, reorganize the figure to stay all on one page. It´s hard to follow as it is now. Also, Maybe try to ensure that the mRNA expression stays in the same column as the WB.

5)    Figure 2: Please, reorganize the figure to stay all on one page.

6)    Figure 5: Are the data expressed as mean ±SE or SD?

7)    Figure 6: Which statistic test was used? Please, include the real p-value.

Results

8)    It would be helpful if the results were presented in an easier format.

9)    Topic 2.4: Did you do any statistic analysis here? If so, please, include it.

10)  Line 225: Please add the p-value. Additionally, include the p-value in the entire text where a statistical analysis was required.

11)  Topic 2.5: I do not understand why to use ANOVA here. Could you explain me, please?

Material and methods

12)  The choice of the statistical tests described is not appropriate. Why ANOVA for 2D, and 3D, for example? Please, review it.  

Discussion

13)  In the discussion, the authors presented the data again. Please, rewrite it.

Reviewer 2 Report

Comments and Suggestions for Authors

Work by Yin-Hua Cheng et al. regarding "A Preliminary Investigation of the Roles of Endometrial Cells 2 in Endometriosis Development via In Vitro and In Vivo Analyses" is an interesting piece of literature dealing with an extremely important medical issue. However, it requires several changes to improve quality and readability:

- in the title in vitro and in vivo it should be written in italics;

- Figure 1 presents too much data, which makes tracking and analysis difficult due to the small size of individual graphs and photos. Please consider dividing it into at least two parts, because in its current state, a lot of important information is simply lost in the multitude of everything. It is also very difficult to decipher which chart has which letter in its designation, some of the letters were left on the previous page; each letter refers to the gel photo and the chart below, which is too confusing.

- Since Figure 1D-O already concerns the part marked in the manuscript as 2.2 Effects of LPS Stimulation on Proliferation, Inflammation, and Fibrosis in Rat ECs In Vitro, why not divide the entire figure into two parts corresponding to individual subsections. It would also be useful to describe the obtained research results in more detail in this section.

-Why do we have expression marked in the context of B-actin in the graphs in Figure 1 and GAPDH in the gels? Why this choice?

- Figure 2 and 3: problems with the letter signature of individual parts of this figure; no scale in the photos from part B;

- the charts in Figures 4 and 5 are very illegible, please improve their quality and size;

- figure 6 is dramatically unreadable, please improve their quality and size; additionally, in the current version it is difficult to compare the results obtained from these two methods, have you considered unifying the scale for individual charts - it would help to assess the differences in these procedures

- the discussion is well written and detailed;

- I would describe procedures 1 and 2 in the materials and methods section in tabular form, which will allow for identifying differences in both approaches.

- in the summary, I lack a stronger emphasis on further research perspectives that should be carried out and in what direction further research on this important issue should go.

Round 2

Reviewer 1 Report

Comments and Suggestions for Authors

The authors conducted a preliminary investigation into the influence of primary endometrial cells (ECs) on the development and progression of endometriosis. The study was well thought out and executed and the major points identified in the first review were addressed.

Reviewer 2 Report

Comments and Suggestions for Authors

Dear authors,

Thank you very much for making changes to the manuscript. In its current form, it is much more readable and easier to read.